# Detecting Recent Dynamics in Large-Scale Landslides via the Digital Image Correlation of Airborne Optic and LiDAR Datasets: Test Sites in South Tyrol (Italy)

Melissa Tondo [1,*], Marco Mulas [1], Giuseppe Ciccarese [1], Gianluca Marcato [2], Giulia Bossi [2], David Tonidandel [3], Volkmar Mair [3] and Alessandro Corsini [1]

1 Department of Chemical and Geological Sciences, University of Modena and Reggio Emilia, Via Giuseppe Campi, 103, 41125 Modena, Italy; marco.mulas@unimore.it (M.M.); giuseppe.ciccarese@unimore.it (G.C.); alessandro.corsini@unimore.it (A.C.)
2 Research Institute for Geo-Hydrological Protection (IRPI), National Research Council (CNR), Corso Stati Uniti, 4, 35127 Padova, Italy; gianluca.marcato@irpi.cnr.it (G.M.); giulia.bossi@irpi.cnr.it (G.B.)
3 Office for Geology and Building Materials Testing, Autonomous Province of Bolzano, Via Val d'Ega, 48, 39053 Cardano, Italy; david.tonidandel@provinz.bz.it (D.T.); volkmar.mair@provincia.bz.it (V.M.)
* Correspondence: melissa.tondo@unimore.it

**Abstract:** Large-scale slow-moving deep-seated landslides are complex and potentially highly damaging phenomena. The detection of their dynamics in terms of displacement rate distribution is therefore a key point to achieve a better understanding of their behavior and support risk management. Due to their large dimensions, ranging from 1.5 to almost 4 $km^2$, in situ monitoring is generally integrated using satellite and airborne remote sensing techniques. In the framework of the *EFRE-FESR SoLoMon* project, three test-sites located in the Autonomous Province of Bolzano (Italy) were selected for testing the possibility of retrieving significant slope displacement data from the analysis of multi-temporal airborne optic and light detection and ranging (LiDAR) surveys with digital image correlation (DIC) algorithms such as normalized cross-correlation (NCC) and phase correlation (PC). The test-sites were selected for a number of reasons: they are relevant in terms of hazard and risk; they are representative of different type of slope movements (earth-slides, deep seated gravitational slope Deformation and rockslides), and different rates of displacement (from few cm/years to some m/years); and they have been mapped and monitored with ground-based systems for many years (DIC results can be validated both qualitatively and quantitatively). Specifically, NCC and PC algorithms were applied to high-resolution (5 to 25 cm/px) airborne optic and LiDAR-derived datasets (such as hillshade and slope maps computed from digital terrain models) acquired during the 2019–2021 period. Qualitative and quantitative validation was performed based on periodic GNSS surveys as well as on manual homologous point tracking. The displacement maps highlight that both DIC algorithms succeed in identifying and quantifying slope movements of multi-pixel magnitude in non-densely vegetated areas, while they struggle to quantify displacement patterns in areas characterized by movements of sub-pixel magnitude, especially if densely vegetated. Nonetheless, in all three landslides, they proved to be able to differentiate stable and active parts at the slope scale, thus representing a useful integration of punctual ground-based monitoring systems.

**Keywords:** large-scale slow-moving landslides; monitoring; GNSS; LiDAR; digital image correlation; Italian Alps

## 1. Introduction

Landslides contribute to the evolution of the Earth's surface as a dominant geomorphic agent, and consequently human-made structures and activities are often directly or indirectly damaged [1,2]. Under the influence of climate change, some types of landslides are expected to increase in intensity and frequency soon, while for others, the effect of

climate change on predisposing and triggering factors still needs to be investigated and understood [3]. Among the first group there are shallow slides, debris flows and rockfalls in high mountain areas, which can be triggered by short and intense rainfall events or by the degradation of permafrost and can also evolve into gully erosion [4]. Among the second group there are large-scale deep-seated landslides, whose time-delayed response to the precipitation regime makes their relationship with climate change more complex and uncertain [5]. In this framework, the detection of the dynamics and evolution at slope scale of large-scale deep-seated landslides can certainly help. In situ Global Navigation Satellite System (GNSS)- and Robotic Total Station-based monitoring systems [6,7] are characterized by high accuracy but limited spatial coverage. Considering landslides covering areas of some square kilometers, the use of remote sensing data, acquired using synthetic aperture radar (SAR) and multispectral satellites or airborne optic and light detection and ranging (LiDAR) surveys, can be extremely useful and informative for the study and interpretation of dynamics occurring in such phenomena [8–10]. Specifically, airborne optic and LiDAR surveys can be used to produce multitemporal high-resolution datasets which are suitable for digital image correlation (DIC), one of the most widely used approaches in the fields of mechanics and optics for measuring deformation [8,9,11–17]. The fundamental principle of DIC is that a couple of geometrically aligned digital images can be "compared", generally with normalized cross correlation (NCC) and phase correlation (PC) algorithms, to assess the deformation that took place in between the two acquisitions [11,12]. In the case of landslides, DIC can allow the assessment of movements at the slope scale with an accuracy which depends on factors such as the adopted DIC algorithms, the digital image resolution, distortion, subset size, etc. [16].

In this work, NCC and PC algorithms are applied to multitemporal (i.e., 2019 and 2021) high-resolution airborne optic and LiDAR datasets of three large-scale landslides located in South Tyrol (Northern Italy). The test sites were selected for their relevance in terms of hazard and risk, and for the fact that they have been extensively studied and monitored with other methods. The availability of other datasets allows the DIC results to be evaluated and validated, in order to assess the feasibility of long-term monitoring via airborne techniques, a goal of the *EFRE-FESR SoLoMon* project (funded by the Autonomous Province of Bolzano-South Tyrol in the framework of the European Regional Development Fund).

## 2. Materials and Methods

### 2.1. Test Sites

The test sites were located in different areas of South Tyrol (Figure 1A): the Corvara landslide in Badia Valley (close to the town of Corvara in Badia); the Ganderberg landslide in Passiria Valley (north of the city of Merano); the Trafoi landslide in Upper Venosta Valley (close to the Stelvio Pass).

The Corvara landslide (Figure 1B) is a large-scale rotational earth slide-earth flow affecting sedimentary rocks consisting of alternations of volcanoclastic sandstones, marls, marly limestones and claystones. It extends over an area of 1.55 km$^2$, ranging in elevation from 1550 m to 2080 m a.s.l. The maximum depth is almost 100 m, and the estimated volume is more than 30 million m$^3$ [18–20]. The landslide body can be divided into source, track, and accumulation zones. The landslide is continuously active, as demonstrated by long-term geotechnical and geomatic monitoring [21–25]. Specifically, GNSS monitoring from 2001 to 2022, carried out as a synergic action between the Office for Geology and Building Materials Testing and the Forestry Department of the Bolzano Autonomous Province, the European Academy of Bolzano (Eurac) and the Chemical and Geological Department of the University of Modena and Reggio Emilia (UniMoRe), has shown a heterogeneous distribution of displacements. It ranges from a few cm/year in the accumulation zone to tens of m/year in parts of the source and track zones [26]. Movements of the Corvara landslide cause damages to the national road SS 244, ski facilities and the local golf course.

The land cover in the landslide area is mostly meadows and bare ground, quite favorable conditions for airborne DIC.

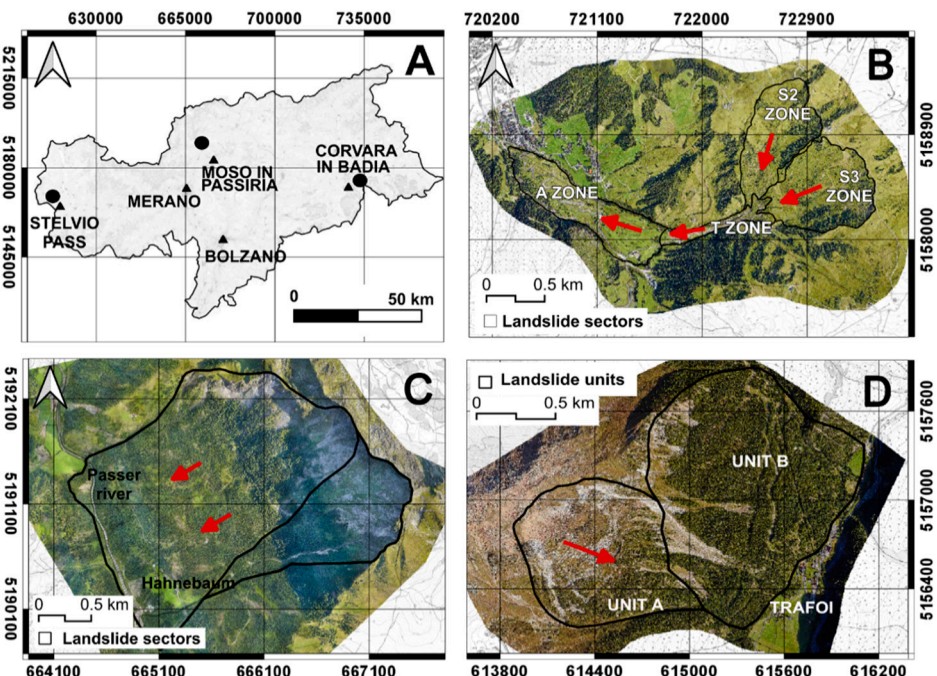

**Figure 1.** (**A**) Location of the test sites in South Tyrol; (**B**) orthophoto of the Corvara landslide, with an outline of the main sectors; (**C**) orthophoto of the Ganderberg landslide with the north-east part not affected by deformation and the north-west one potentially influenced by rock failures; (**D**) orthophoto of the Trafoi landslide with an outline of the two main units. Red arrows indicate the main direction of displacement. Coordinate system: WGS84 32N.

The Ganderberg landslide (Figure 1C) is a deep-seated gravitational slope deformation (DSGSD) affecting metamorphic rocks such as paragneiss, mica-schist, and marbles. It extends over an area of 3.75 km$^2$ with an estimated maximum depth of 100–200 m, ranging in elevation from 1170 m to 2330 m a.s.l. The gravitational collapse of the slope is generating a tension field along the crown where a rock slab of about 800.000 m$^3$ displays traces of potential detachment [27]. The morphology of the landslide body is typical of a DSGSD in its final stage, with a convex curvature on the upper part, flat in the center and concave at the toe. The low quality of the bedrock in conjunction with the continuous movements of the DSGSD determines the weakening of the involved rock masses, leading to collateral phenomena such as debris flow, avalanches, or secondary slides [28]. In particular, the north-east part of the DSGSD crown, consisting of mica-schists, is not significantly affected by deformations, while the north-west part, consisting of steep slopes of paragneiss, shows transverse cracks and joints that can potentially release massive rock blocks [27]. In medieval times, in 1401, a rock avalanche triggered by the collapse of a rock slab from the north-west ridge of Mount Ganderberg led to the formation of a natural dam on the Passer River whose breaching, in 1419, caused flooding and 400 casualties downstream in the city of Merano [28]. Periodic GNSS monitoring (carried out by the Forestry Department of the Bolzano Autonomous Province, IRPI-CNR of Padua and Helica S.r.l.) during the 2007–2021 period has shown that the DSGSD is characterized by an almost constant velocity of 6–8 cm/year, causing damages to the national road SS 44bis and the hamlet of Hahnebaum [27]. Land cover, which represents a challenge for DIC application, is mostly characterized by woods and some limited meadows, while bare ground, which is better for this application, is visible on scree slopes only.

The Trafoi landslide (Figure 1D) is a DSGSD evolving into a deep-seated rockslide, affecting metamorphic rocks such as paragneiss and orthogneiss. It extends over an area of

2.5 km$^2$ ranging in elevation from 1550 m to 2700 m a.s.l. The maximum depth is unknown, but given the geometry and characteristics of the phenomenon it is expected to be more than 150 m. This leads to an estimated volume of more than 40 million m$^3$. The landslide can be subdivided into two main units. Unit A (1.2 km$^2$) comprises the main deep-seated rockslide that is located in the central part of the slope (where the rock mass is highly fractured, generating also abundant material for debris flows) and the uppermost DSGSD, which causes double crested ridges and trenches [29,30]. Unit B (1.3 km$^2$) comprises a more evolved rockslide deposit which locally evolves into earth slides. Periodic GNSS monitoring (carried out by the Forestry Department of the Autonomous Province of Bolzano and Helica S.r.l.) of Unit A during the 2008–2021 period has shown that the deep-seated rockslide moves at a rate of 1.5 to 6.5 cm/year while the DSGSD is much slower, less than 1 cm/year. Movements of the rockslide are worthy of attention since further acceleration might lead the phenomenon to evolve into a rock avalanche, an occurrence that would have catastrophic consequences for the village of Trafoi. At the same time, inclinometers have shown that Unit B moves locally at a few cm/year. In this portion there is no potential for a catastrophic evolution, but nonetheless such movements have repeatedly damaged a hotel and the national road SS 38 [29,30]. Land cover is bare ground on scree slopes and most of the Unit A rockslide, determining a quite favorable condition for airborne DIC. On the contrary, Unit B is mostly covered by woodland and some patches of meadows, resulting in challenging conditions for airborne DIC.

### 2.2. Materials

In all three case studies, airborne surveys with a high-resolution optical camera and a long-range multi-pulse LiDAR device were carried out in 2019, 2020, and 2021 under snow and cloud-free conditions by the company Helica s.r.l. They extracted the LiDAR point clouds with RIPROCESS 1.8.7 software, then filtered and classified these points using Terrasolid 2020 Package into 4 classes: ground, over ground, low and air points; then, full-resolution digital surface model tiles (fr-DSM) and full-resolution digital terrain model tiles (fr-DTM) were elaborated for each site and each survey. A mean distribution of 20 ground pts/m$^2$ for the Corvara and Ganderberg landslides, and 12 ground pts/m$^2$ for Trafoi landslide, was calculated by Helica s.r.l. Optical images were radiometrically corrected and orthorectified using Agisoft Photoscan software building up full-resolution digital Ortho-Photos tiles (fr-OPH). For extent, tiling and resolution details of the original datasets, see Table 1.

**Table 1.** Characteristics of original datasets produced for each test site by Helica S.r.l.

| Site | Date | Extent | Original Data | Format | Resolution |
|---|---|---|---|---|---|
| Corvara | 10, 14 August 2019<br>30 July 2020<br>24 August 2021 | 6.5 km$^2$<br>(34 tiles) | fr-OPH<br>fr-DSM<br>fr-DTM | tiff<br>ascii<br>ascii | 0.05 m/pixel<br>0.25 m/pixel<br>0.05 m/pixel |
| Ganderberg | 15, 16 August 2019<br>29 July 2020<br>23, 24 September 2021 | 11.6 km$^2$<br>(66 tiles) | fr-OPH<br>fr-DSM<br>fr-DTM | tiff<br>ascii<br>ascii | 0.05 m/pixel<br>0.25 m/pixel<br>0.05 m/pixel |
| Trafoi | 14 August 2019<br>28 July 2020<br>23 September 2021 | 4.3 km$^2$<br>(29 tiles) | fr-OPH<br>fr-DSM<br>fr-DTM | tiff<br>ascii<br>ascii | 0.05 m/pixel<br>0.25 m/pixel<br>0.05 m/pixel |

### 2.3. Methods

DIC is an optical-numerical measurement technique that can be employed to find and quantify movements through multitemporal digital imagery [16]. This technique implements algorithms that geometrically align two or more images of an area (the older image defined as the master and the more recent one defined as the slave) obtained during different periods [11]. Displacements are calculated by assessing the internal

misalignment through a moving template window that compares master and slave images taken before and after a certain deformation event. DIC usually allows the measurement of displacements without the installation of any sensors or reflectors in the investigated area; thus, it can be considered as a fully remote measurement system [16].

Identifying a perfect correspondence between single pixels in master and slave images is impossible, as the value in one pixel can be equal to those of thousands of other pixels. Comparison between images is therefore obtained considering a pixel and a discrete area around it [11,12]. The wideness of this area is selected by the operator, and it represents the analysis mask in which the DIC algorithms operate. The subset size must be carefully chosen, and it should not either be too small or too large as it cannot describe small or large heterogeneous deformations, respectively, and will affect the sub-pixel accuracy. In other terms, the subset size strongly depends on the characteristics of the images. In addition, another important parameter that must be selected thoroughly is the step size, which represents the shift of the subset during the correlation process [11,12,16].

Among existing algorithms for DIC, quite commonly adopted ones are NCC and PC. NCC operates in the spatial domain, by searching for similar-intensity contrasts between a square template window from the master image and a square search window from the slave image. Template and search windows move simultaneously, shifting horizontally over the master and slave images, thus enabling offset estimates [16]. PC operates in the frequency domain, thus searching for phase differences between master and slave images, and it is based on the application of Fast Fourier transforms (FFT) to estimate translative offset between them [16,31,32]. In this work, NCC and PC were applied using IRIS software [17,33–35]. IRIS is a commercial software developed by Nhazca S.r.l. (spin-off of the La Sapienza University of Rome) that implements different DIC algorithms and is equipped with a graphical user interface that guides users through modules concerning pre-processing, displacement analysis and post-processing (Figure 2).

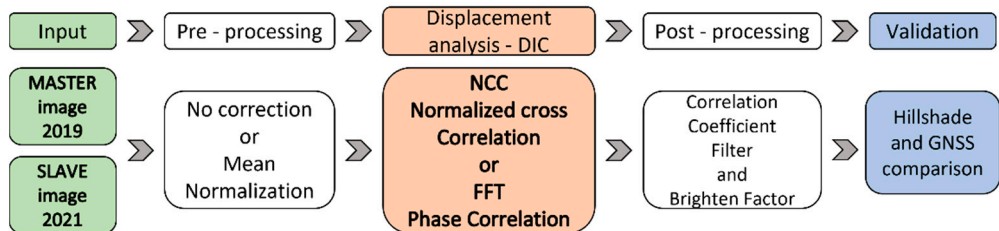

**Figure 2.** Scheme representing the steps adopted for the DIC analysis through IRIS software developed by Nhazca S.r.l.

The following derivative datasets were calculated and further tiled using Q-GIS and IRIS software (see Table 2 for details) to be able to perform DIC analysis on a low-end workstation (Intel i5 CPU 3.20 GHz, 8 GB RAM): (i) lowered-resolution digital RGB orthophotos (OPH); (ii) lower/full-resolution hillshade models (HSD); (iii) lower/full-resolution slope angle models (SLOPE). The RGB OPH datasets were resampled in Q-GIS and pre-processed in IRIS via mean normalization with a window size of 10 pixels and a step size of 2 pixels. Within IRIS software, pre-processing of orthophotos was required to improve the final results, while no pre-processing was necessary for hillshade and slope models.

In order to assess which combination of dataset vs. algorithm can lead to a better detection and quantification of slope movements, in the Corvara landslide, which has movement rates up to m/year and quite favorable land cover conditions, DIC analysis took all available datasets (OPH, HSD, SLOPE) into consideration. On the other hand, for the case studies of Ganderberg and Trafoi, which are characterized by movement rates in the order of only a few cm/years, the DIC analysis focused solely on SLOPE datasets at different resolution.

**Table 2.** Characteristics of derivative datasets used in each test site for DIC analysis.

| Site | Year | Extent | Tiles | Derivative Data | Resolution |
|---|---|---|---|---|---|
| Corvara | 2019 and 2021 (two-years interval) | 6.5 km$^2$ | 11<br>3<br>3 | OPH<br>HSD<br>SLOPE | 0.10 m/pixel<br>0.25 m/pixel<br>0.25 m/pixel |
| Ganderberg | 2019 and 2021 (two-years interval) | 6 km$^2$ | 35<br>4 | SLOPE-1<br>SLOPE-2 | 0.05 m/pixel<br>0.25 m/pixel |
| Trafoi | 2019 and 2021 (two-years interval) | 1.4 km$^2$ | 9<br>2 | SLOPE-1<br>SLOPE-2 | 0.05 m/pixel<br>0.25 m/pixel |

This choice was based on the following considerations: (1) land cover in Ganderberg and, to a lesser extent, in Trafoi, mostly consists of woods, making the use of orthophotos problematic. (2) Slope maps show a lower frequency of artifacts, which increase mismatches in DIC analysis. (3) The limited movement rates expected in these landslides should be recognizable more easily with datasets at a higher resolution (i.e., slope maps with 5 cm/pixels resolution). Running the NCC and PC algorithms with IRIS software required defining processing settings tailored to the specific characteristics of the test sites (i.e., expected movement rates) and to the resolution of the datasets under analysis. A detailed list of the adopted processing settings is presented in Table 3, which includes search window size, template window size (for NCC), apodization radius-search radius, subpixel resolution, and pyramid levels. Finally, post-processing consisted of the application of a "Correlation Coefficient Filter", to filter off pixels with too low cross correlation values, and of a "Brighten Factor", to enhance the brightness of the resulting displacement maps (see Table 3 for values).

**Table 3.** List of processing settings implemented in the DIC analyses.

| Site | DIC Method-Dataset | Search Window (px) | Template Window (px) | Apodization Radius-Search Radius (px) | Subpixel Resolution (px) | Pyramid Levels | Correlation Coefficient Filter | Brighten Factor |
|---|---|---|---|---|---|---|---|---|
| Corvara | PC–OPH | 32 | - | 0.30 | 0.05 | 3 | ≤0.05 | 4 |
| | NCC–OPH | 32 | 16 | 1 | 0.05 | 4 | ≤0.05 | 4 |
| | PC–HSD | 32 | - | 0.30 | 0.05 | 3 | ≤0.05 | 4 |
| | NCC–HSD | 64 | 32 | 1 | 0.5 | 4 | ≤0.05 | 4 |
| | PC–SLOPE | 32 | - | 0.30 | 0.05 | 3 | ≤0.05 | 4 |
| | NCC–SLOPE | 64 | 32 | 1 | 0.5 | 4 | ≤0.05 | 4 |
| Ganderberg | PC–SLOPE-1 | 8 | - | 0.30 | 0.5 | 3 | ≤0.10 | 4 |
| | NCC–SLOPE-1 | 128 | 64 | 1 | 0.05 | 4 | ≤0.20 | 7 |
| | PC–SLOPE-2 | 32 | - | 0.30 | 0.05 | 3 | ≤0.05 | 4 |
| | NCC–SLOPE-2 | 128 | 64 | 1 | 0.05 | 4 | ≤0.20 | 7 |
| Trafoi | PC–SLOPE-1 | 8 | - | 0.30 | 0.5 | 3 | ≤0.10 | 4 |
| | NCC–SLOPE-1 | 128 | 64 | 1 | 0.05 | 4 | ≤0.20 | 7 |
| | PC–SLOPE-2 | 32 | - | 0.30 | 0.05 | 3 | ≤0.05 | 4 |
| | NCC–SLOPE-2 | 128 | 64 | 1 | 0.05 | 4 | ≤0.20 | 7 |

For quantitative validation purposes, the DIC results were compared to slope movement data obtained via GNSS monitoring of permanent ground benchmarks, with periodic measurement campaigns carried out almost simultaneously with airborne surveys (see Table 4 for details). Furthermore, for each test-site, the DIC results have been compared to

displacement values estimated by visual homologous points tracking (HPT), i.e., by means of operator-based identification and digitization of the changing position of homologous ground features (HGF) clearly recognizable in the HSD of 2019 and 2021. In addition, the adjusted R-squared ($R^2$) and root mean squared error (RMSE) were calculated in the *R* environment [36] to assess the accuracy and precision of each application.

**Table 4.** List of datasets from GNSS campaigns used for validation.

| Site | Month/Year | Operator | Technique | Benchmarks (GNSS) |
|---|---|---|---|---|
| Corvara | 10/2019, 11/2020; 09/2021 | Forestry Department | Fast static | 22 |
| Ganderberg | 10–11/2019, 07/2020, 09/2021 | Helica S.r.l. | Fast static and RTK | 23 |
| Trafoi | 10–11/2019, 07/2020, 09/2021 | Helica S.r.l. | Fast static and RTK | 11 |

## 3. Results

In the following section, displacement maps obtained with DIC analysis through the IRIS software will be described, together with the validation methods implemented to assess the capacity of this technique to detect and quantify movements. According to Template Window dimensions, DIC analysis on a couple of images is always affected by a no-data peripheral frame. The elaborated products show no-data strips which are a consequence of the tile subdivision. Therefore, in the next images in this section, grey lines may attract the attention of the reader, but they merely are the combination of such no-data strips and the hillshade below. Regarding the validation methods, the movement values extracted from all displacement maps correspond to a median of values falling within a buffer circled area with a radius of 3 m, to minimize the outliers around both HPT and GNSS points.

### 3.1. Corvara Landside

Results for the Corvara landslide are presented in Figure 3 and refer to slope movements in a two-year period (2019 to 2021) assessed on the basis of: (i) orthophotos at a 10 cm/px resolution with Phase Correlation (PC-OPH, Figure 3A) and normalized cross correlation (NCC-OPH, Figure 3B); (ii) hillshade at a 25 cm/px resolution with phase correlation (PC-HSD, Figure 3C) and normalized cross correlation (NCC-HSD, Figure 3D); (iii) slope map at a 25 cm/px resolution with phase correlation (PC-Slope, Figure 3E) and normalized cross-correlation (NCC-Slope, Figure 3F). These results were later validated using long-term GNSS measurements (Figure 3G) and operator-based identification and measurement of the changing position of recognizable HGF, as shown in Figure 3H.

Displacement maps highlight, quite similarly to one another, slope movements of variable magnitude inside the S2 zone, S3 zone, and the upper T zone. These are consistent with field geomorphic evidence and independent long-term monitoring data, which have identified these portions of the slope as the most active ones, with seasonal movements that can reach the magnitude of several m/year [21,24,25]. Nevertheless, the detection capacity and the quantification of movements change with changing datasets and algorithms, as well as with the cumulated displacement reached during the test period. It appears that the use of HSD and SLOPE datasets enhances the movement detection capacity in the application of both NCC and PC algorithms as the resulting displacement maps are, in fact, consistent with geomorphic evidence.

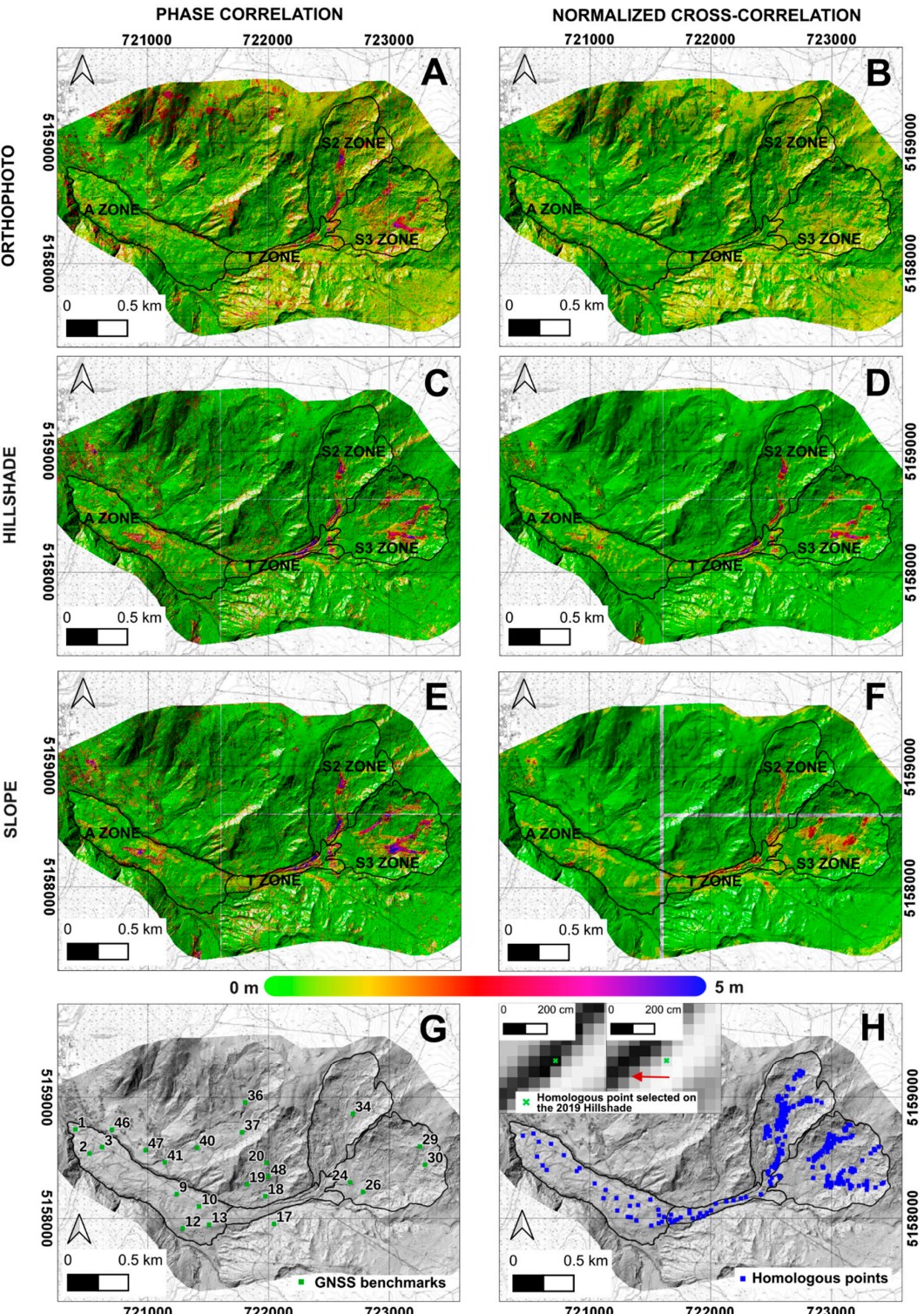

**Figure 3.** Displacement maps related to the evolution of the Corvara landslide during the 2019–2021 period. (**A**) PC-OPH. (**B**) NCC-OPH. (**C**) PC-HSD. (**D**) NCC-HSD. (**E**) PC-SLOPE. (**F**) NCC-SLOPE. (**G**) Distribution of GNSS benchmarks measured by the Forestry Department during the 2019–2021 period. (**H**) Position of homologous ground features recognizable on the hillshade of 2019 and 2021. Red arrow indicates the movement direction (analysis carried out using IRIS software, developed by Nhazca S.r.l.). Coordinate system: WGS84 32N.

On the other hand, OPH datasets, despite the high resolution, proved to be the least effective for the analysis of movements, especially with NCC. With respect to the quantitative assessment of movements, PC tends to compute larger movements than NCC with any of the adopted datasets. For instance, as shown in Figure 4, movement rates of 94 HGF located in the S2 zone calculated with PC mainly range between 0.5 m and 4 m, while movements rates calculated with NCC range mainly between 0.5 m and 2.5 m (Figure 4A–C). The same pattern can be recognized in the plots representing movement rates calculated in 25 HGF along the track zone (Figure 4D–F). Overall, compared to HPT values which mainly reach 2–3 m, both algorithms tend to overestimate movements. Despite this, the best correlations are obtained with NCC application (Figure 4B,E). As previously stated, another assessment of DIC results was conducted comparing the calculated displacements with 22 GNSS benchmarks. In general, quantitative correlations are poor for any kind of algorithm–dataset combinations: GNSS measurements range from 0 to almost 0.7 m while DIC results mainly range between 0.1 m and 1.2 m, retaining the tendency of overestimating displacements. Nevertheless, among the three plots in Figure 5, the best correlation is reached firstly with the SLOPE dataset (Figure 5C) and secondly with the HSD one (Figure 5B).

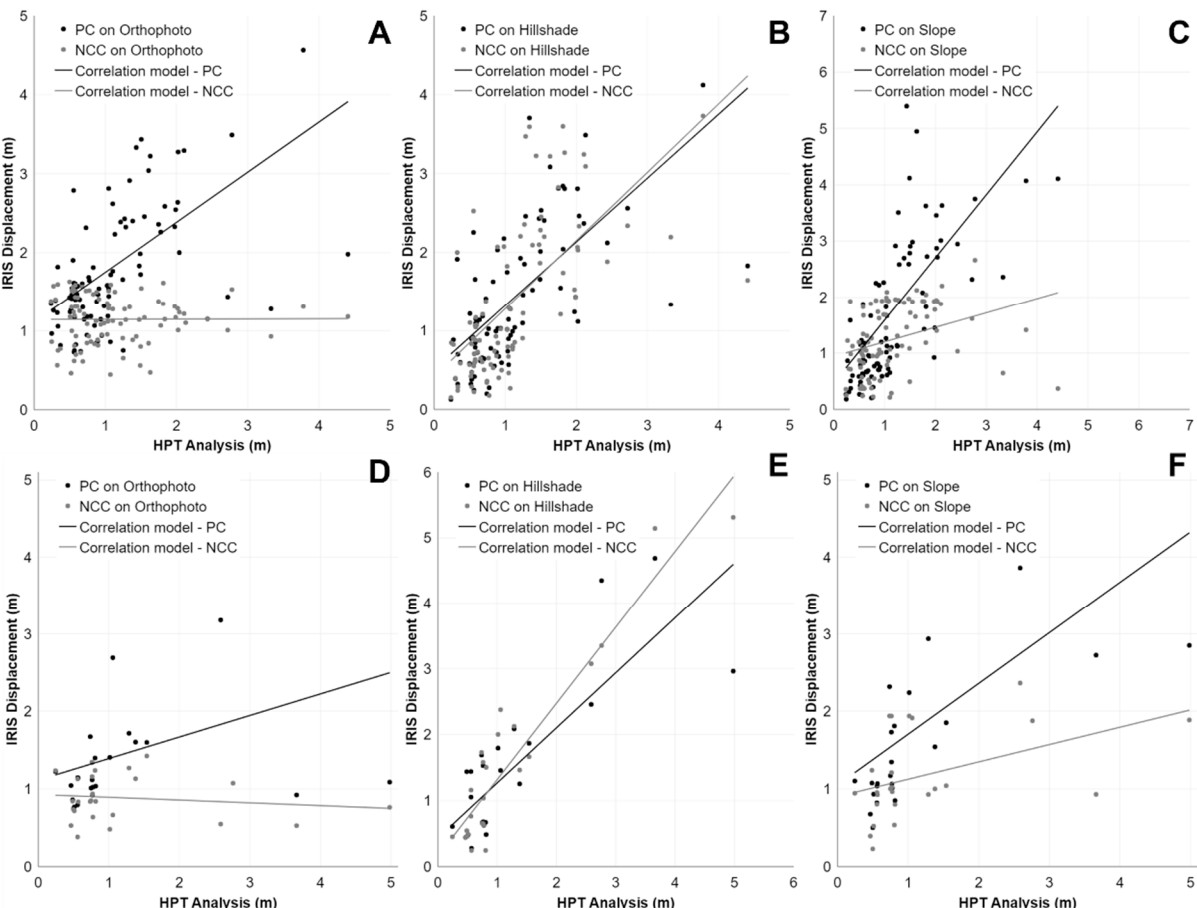

**Figure 4.** Scatter plots representing the correlation between PC, NCC and measurements of homologous points (HPT analysis) with different datasets in the Corvara S2 zone (**A**–**C**) and in the T zone (**D**–**F**). (**A**) PC-OPH ($R^2$: 0.3, RMSE: 1.03); NCC-OPH ($R^2$: −0.011, RMSE: 0.81). (**B**) PC-HSD ($R^2$: 0.39, RMSE: 0.82). NCC-HSD ($R^2$: 0.41, RMSE: 0.83) (**C**) PC-SLOPE ($R^2$: 0.32, RMSE: 1.3); NCC-SLOPE ($R^2$: 0.1, RMSE: 0.8). (**D**) PC -OPH ($R^2$: 0.08, RMSE: 1.17); NCC-OPH ($R^2$: −0.02, RMSE: 1.22) (**E**) PC-HSD ($R^2$: 0.65, RMSE: 0.72); NCC-HSD ($R^2$: 0.87, RMSE: 0.61). (**F**) PC-SLOPE ($R^2$: 0.34, RMSE: 1.19); NCC-SLOPE ($R^2$: 0.16, RMSE: 0.99).

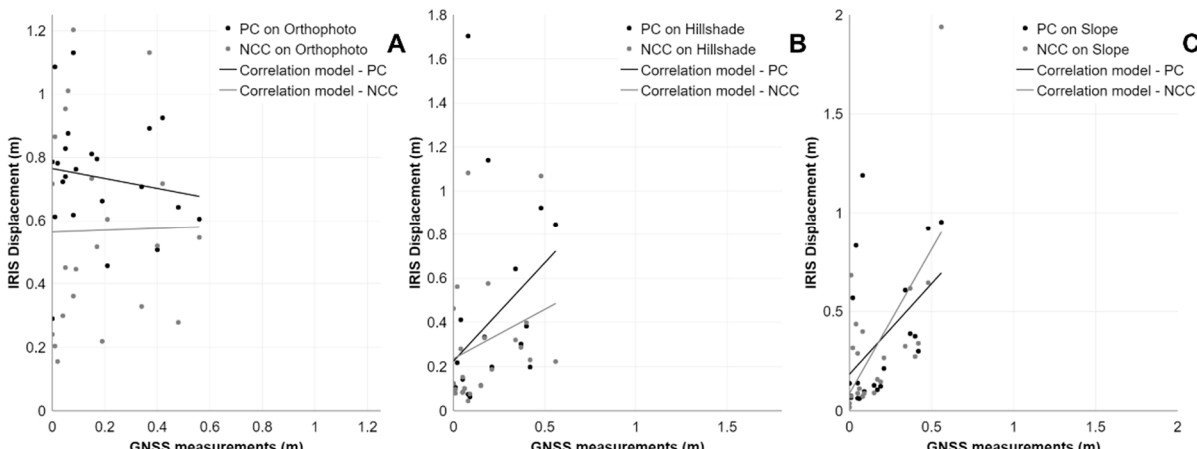

**Figure 5.** Scatter plots representing the correlation between PC, NCC, and GNSS measurements with different datasets in the Corvara landslide. (**A**) PC-OPH ($R^2$: $-0.05$, RMSE: 0.53); NCC-OPH ($R^2$: $-0.03$, RMSE: 0.63). (**B**) PC-HSD ($R^2$: 0.09, RMSE: 0.44); NCC-HSD ($R^2$: 0.03, RMSE: 0.32). (**C**) PC-SLOPE ($R^2$: 0.17, RMSE: 0.35); NCC-SLOPE ($R^2$: 0.36, RMSE: 0.36).

Considering the morphologic, kinematic and land cover conditions characterizing the Corvara landslide, a better detection and quantification performance is obtained with HSD and SLOPE datasets. As for the algorithms, they both show inaccuracies, especially regarding those parts of the landslide body covered with vegetation and the residential areas, but, in general, the application of NCC proved to be the most consistent and smooth method for this type of phenomenon.

### 3.2. Ganderberg Landside

Results for the Ganderberg landslide are presented in Figure 6 and refer to slope movements in a two-year period (2019 to 2021) assessed on the basis of: (i) slope models at a 5 cm/px resolution with phase correlation (PC-SLOPE-1, Figure 6A) and normalized cross-correlation (NCC-SLOPE-1, Figure 6B); (ii) slope models at a 25 cm/px resolution with phase correlation (PC-SLOPE-2, Figure 6C) and normalized cross-correlation (NCC-SLOPE-2, Figure 6D). In Ganderberg, results differ significantly from each other, both in terms of areas identified, or not, as moving, and in terms of computed magnitude of movements. With respect to qualitative displacement detection, the NCC algorithm returns a somewhat more reasonable overall picture of the moving slope, both with high-resolution (i.e., 5 cm/px) and lower-resolution (i.e., 25 cm/px) datasets. However, it also detects an unrealistically large number of scattered pixels with non-negligible movements (Figure 6B) when used with high-resolution datasets (i.e., 5 cm/px), while with the lower-resolution dataset (Figure 6D), it evidences movements mostly in the upper scree-slopes. Nevertheless, in comparison, PC performs relatively worse in movement detection with both the high- and the lower-resolution datasets. In fact, it unreasonably underestimates the extent of moving areas with high-resolution data (Figure 6A) and, on the contrary, largely overestimates their extent with low-resolution data (Figure 6C). Regarding the quantitative assessment of movements, similarly to the Corvara landslide, a comparison with supervised measurements of 68 HGF easily recognizable in both 2019 and 2021 hillshade (Figure 6F) together with 23 GNSS benchmarks (Figure 6E) monitored from 2019 to 2021 was carried out. The results of this comparison, which can be observed in Figures 7 and 8, show that PC and NCC do not have a good correlation with either HPT analysis (Figure 7A,B) or GNSS measurements (Figure 8A,B). Contrarily to the Corvara landslide, in this case, NCC tends to overestimate while PC tends to underestimate the displacement magnitude. Moreover, quantitatively, PC performs better than NCC with both low- and high-resolution datasets, even though NCC proved to be more efficient in qualitatively detecting active areas.

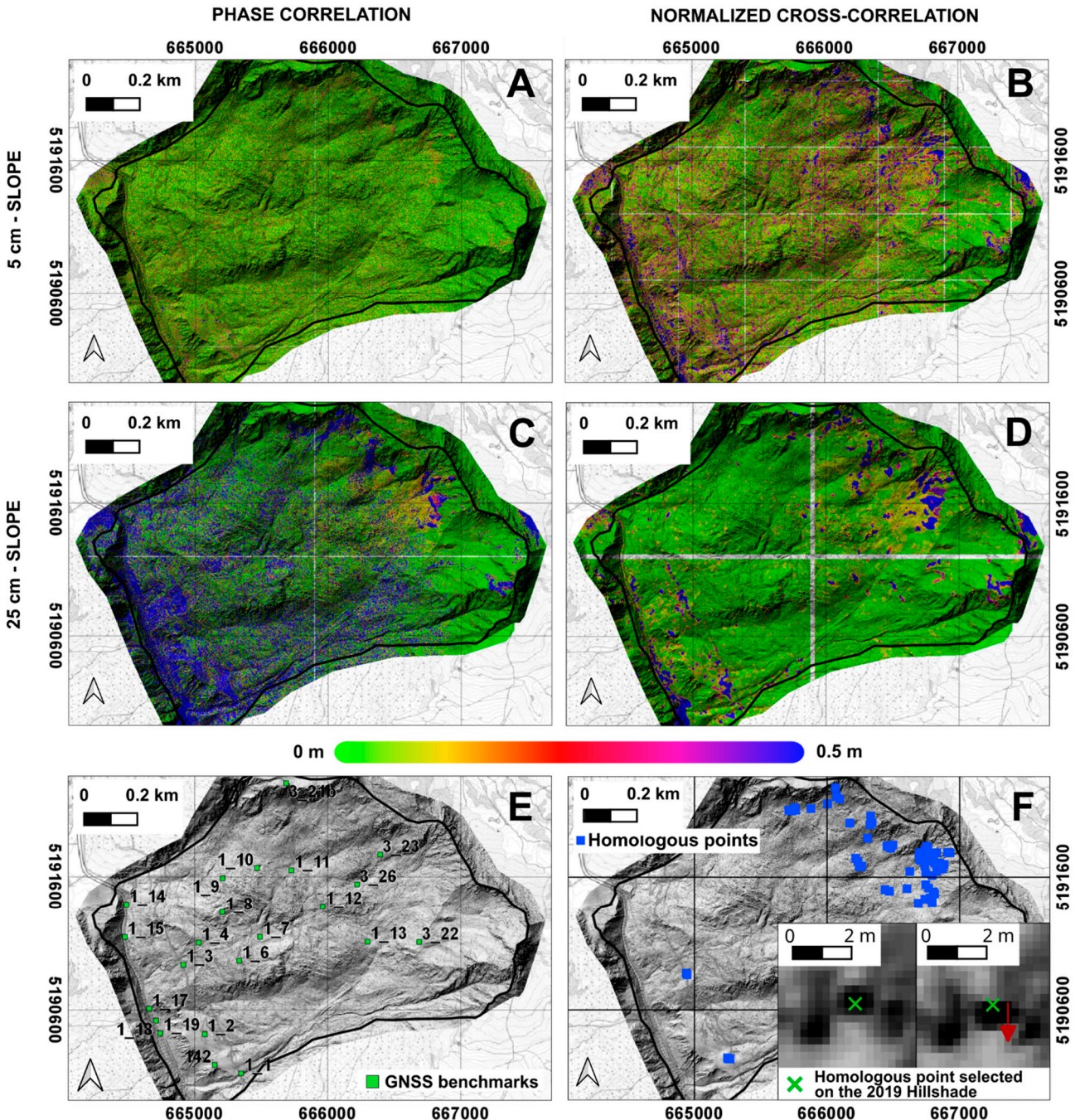

**Figure 6.** Displacement maps related to the evolution of the Ganderberg landslide during the 2019–2021 period. (**A**) PC-SLOPE-1. (**B**) NCC-SLOPE-1. (**C**) PC-SLOPE-2. (**D**) NCC-SLOPE-2. (**E**) Distribution of GNSS benchmarks measured by Helica S.r.l. during the 2019–2021 period. (**F**) Position of homologous ground features recognizable on hillshade of 2019 and 2021. Red arrow indicates the movement direction (analysis carried out using IRIS software, developed by Nhazca S.r.l.). Coordinate system: WGS84 32N.



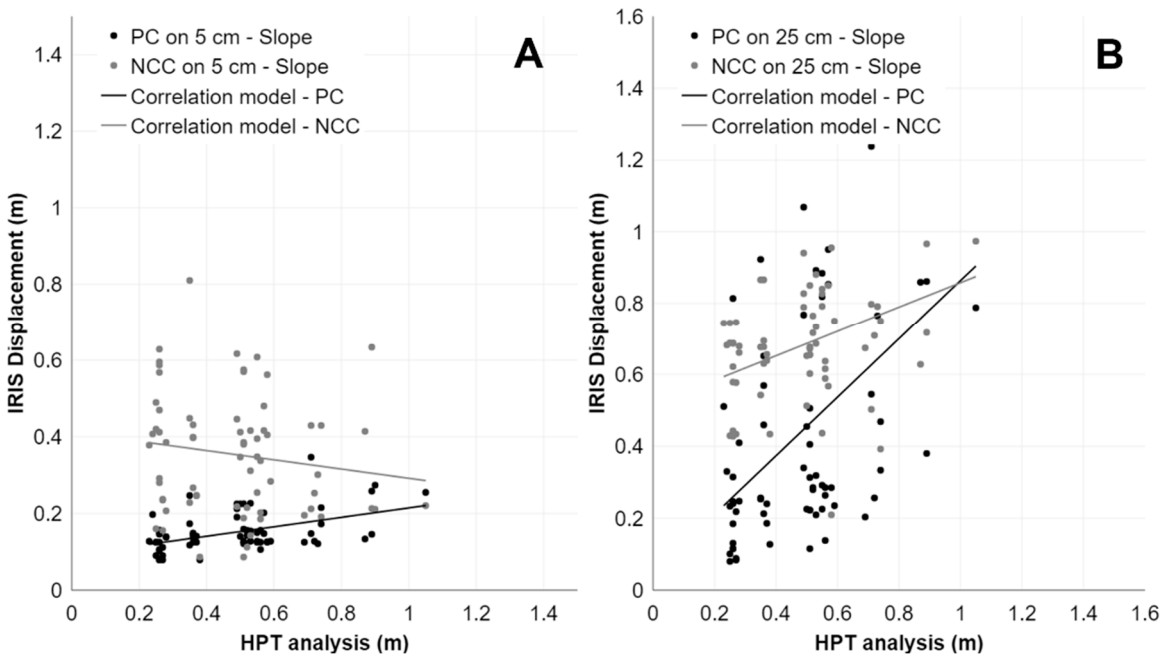

**Figure 7.** Scatter plots representing the correlation between PC and NCC applied on Ganderberg SLOPE and measurements of homologous points (HPT analysis) with different dataset resolutions. (**A**) PC-SLOPE-1 ($R^2$: 0.21, RMSE: 0.37); NCC-SLOPE-1 ($R^2$: 0.01, RMSE: 0.30). (**B**) PC-SLOPE-2 ($R^2$: 0.20, RMSE: 0.31); NCC-SLOPE-2 ($R^2$: 0.14, RMSE: 0.29).

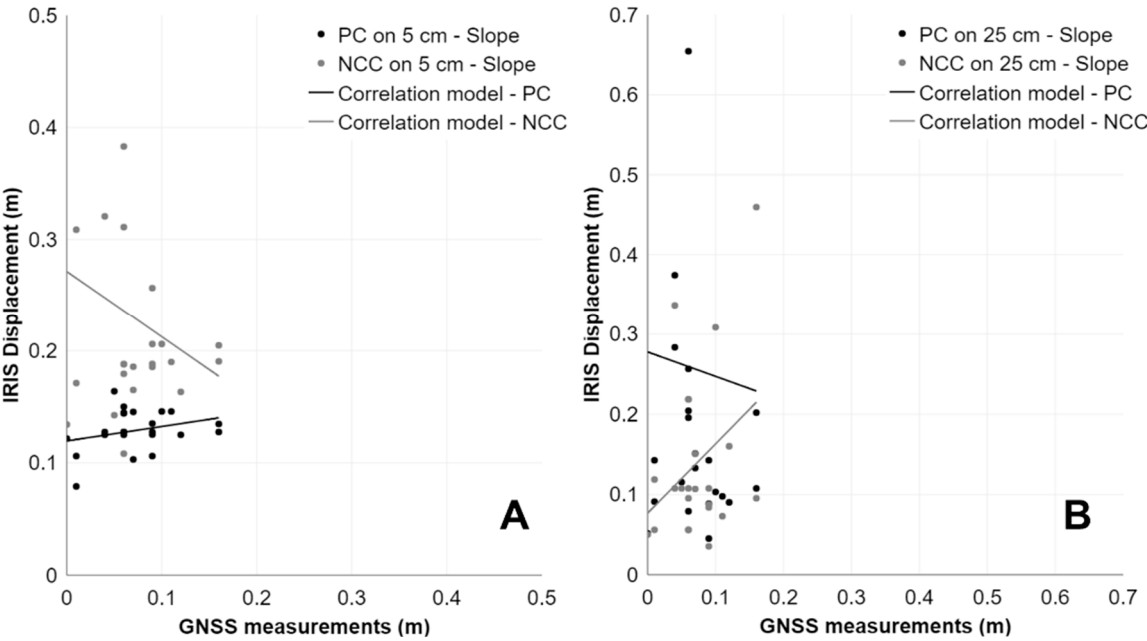

**Figure 8.** Scatter plots representing the correlation between PC and NCC applied on Ganderberg SLOPE and GNSS measurements with different dataset resolutions. (**A**) PC-SLOPE-1 ($R^2$: 0.04, RMSE: 0.07); NCC-SLOPE-1 ($R^2$: −0.004, RMSE: 0.20). (**B**) PC-SLOPE-2 ($R^2$: −0.04, RMSE: 0.12); NCC-SLOPE-2 ($R^2$: 0.07, RMSE: 0.20).

### 3.3. Trafoi Landside

Results for the Trafoi landslide are presented in Figure 9 and refer to slope movements in a two-year period (2019 to 2021) assessed on the basis of: (i) slope models at a 5 cm/px resolution with phase correlation (PC-SLOPE-1, Figure 9A) and the normalized cross-

correlation algorithm (NCC-SLOPE-1, Figure 9B); slope models at a 25 cm/px resolution with phase correlation (PC-SLOPE-2, Figure 9C) and the normalized cross-correlation algorithm (NCC-SLOPE-2, Figure 9D). Like the Ganderberg landslide, the results in Trafoi provide quite variable estimates of slope movements in the different parts of the landslide with different datasets and methods, which are only partially consistent with geomorphic evidence and independent long-term monitoring data. With respect to movement detection, the NCC algorithm returns the more reasonable overall picture of movement distribution, both with high-resolution (i.e., 5 cm/px) and lower-resolution (i.e., 25 cm/px) datasets.

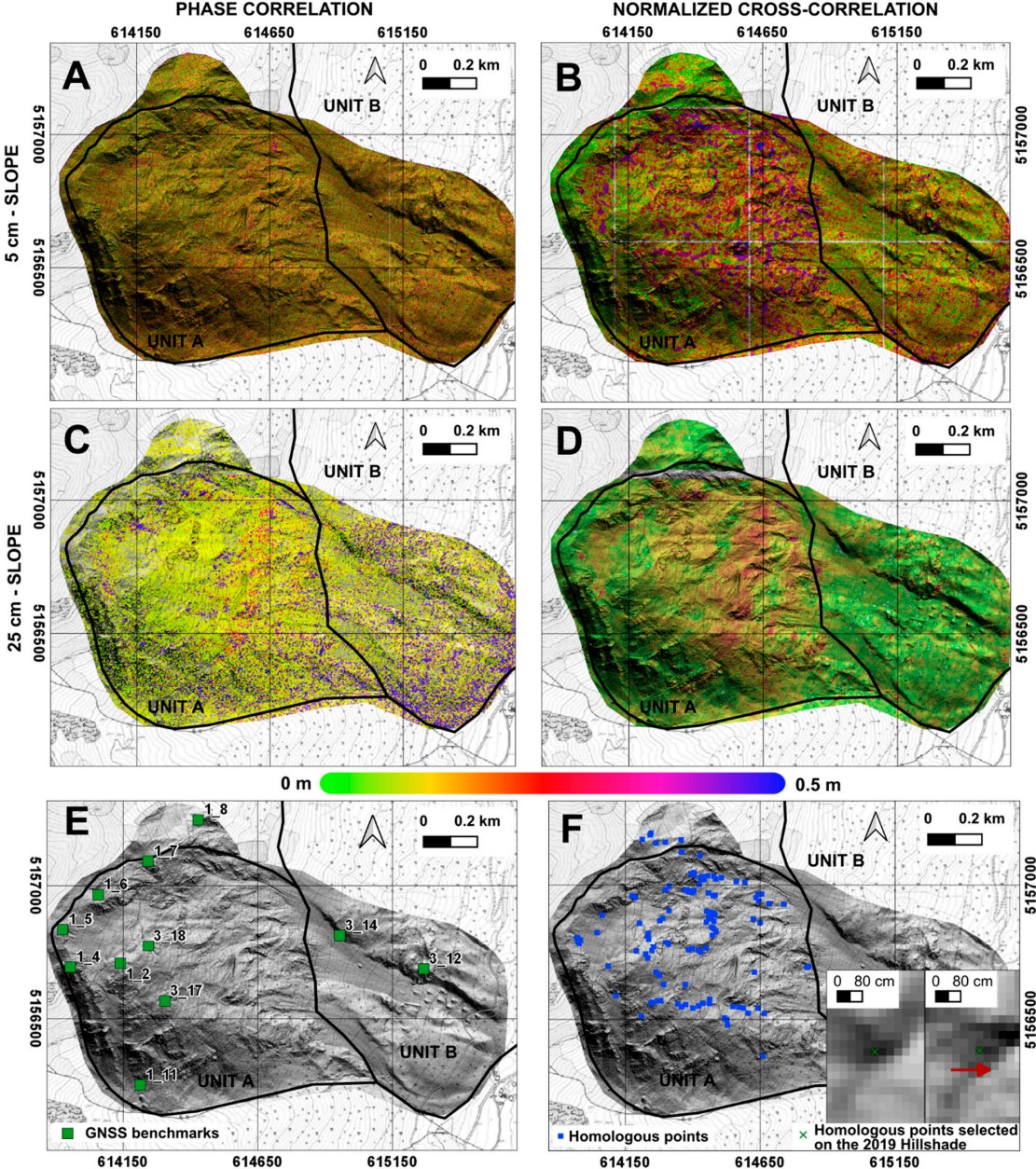

**Figure 9.** Displacement maps related to the evolution of the Trafoi landslide during the 2019–2021 period. (**A**) PC-SLOPE-1. (**B**) NCC-SLOPE-1 (**C**) PC-SLOPE-2. (**D**) NCC-SLOPE-2. (**E**) Distribution of GNSS benchmarks measured by Helica S.r.l. during the 2019–2021 period. (**F**) Position of homologous ground features recognizable on hillshade of 2019 and 2021. Red arrow indicates the movement direction (analysis carried out using IRIS software, developed by Nhazca S.r.l.). Coordinate system: WGS84 32N.

Compared to the Ganderberg maps, the results in Trafoi are somewhat of a better quality, since the pixel values obtained with NCC applied to high-resolution datasets (Figure 9B), although quite scattered, nicely outline the extent of the active rockslide in Unit A and differ significantly from values obtained in the stable areas around it. Furthermore, even with the lower-resolution dataset (Figure 9D), the overall movement of the rockslide is still detected by NCC. In addition, similarly to Ganderberg, the PC performs relatively, but significantly, worse. It unreasonably underestimates the extent of moving areas when applied to high-resolution datasets (Figure 9A) and it returns an undistinguished scattered pixelated picture of movements when applied to lower-resolution datasets (Figure 9C).

As for the quantitative assessment of movements, like in the other test sites, a comparison with supervised measurements of 113 HGF easily recognizable in 2019 and 2021 hillshade (Figure 9F) together with 11 GNSS benchmarks (Figure 9E) monitored from 2019 to 2021 was carried out. The results of this comparison, which can be observed in Figures 10 and 11, show that PC and NCC do not have a good correlation with either HPT analysis (Figure 10A,B) or GNSS measurements (Figure 11A,B).

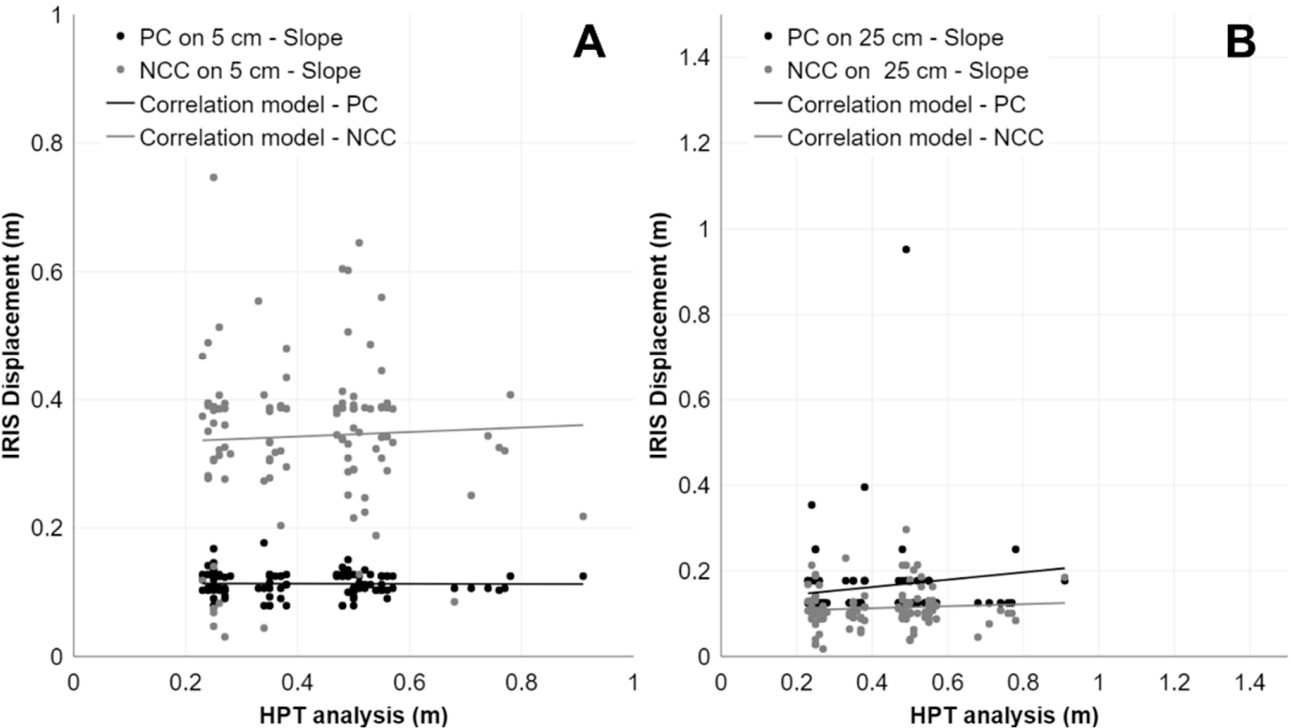

**Figure 10.** Scatter plots representing the correlation between PC and NCC applied on Trafoi SLOPE and measurements of homologous points (HPT analysis) with different resolutions. (**A**) PC-SLOPE-1 ($R^2$: $-0.009$, RMSE: 0.33); NCC-SLOPE-1 ($R^2$: $-0.007$, RMSE: 0.20). (**B**) PC-SLOPE-2 ($R^2$: $-0.001$, RMSE: 0.32); NCC-SLOPE-2 ($R^2$: $-0.002$, RMSE: 0.34).

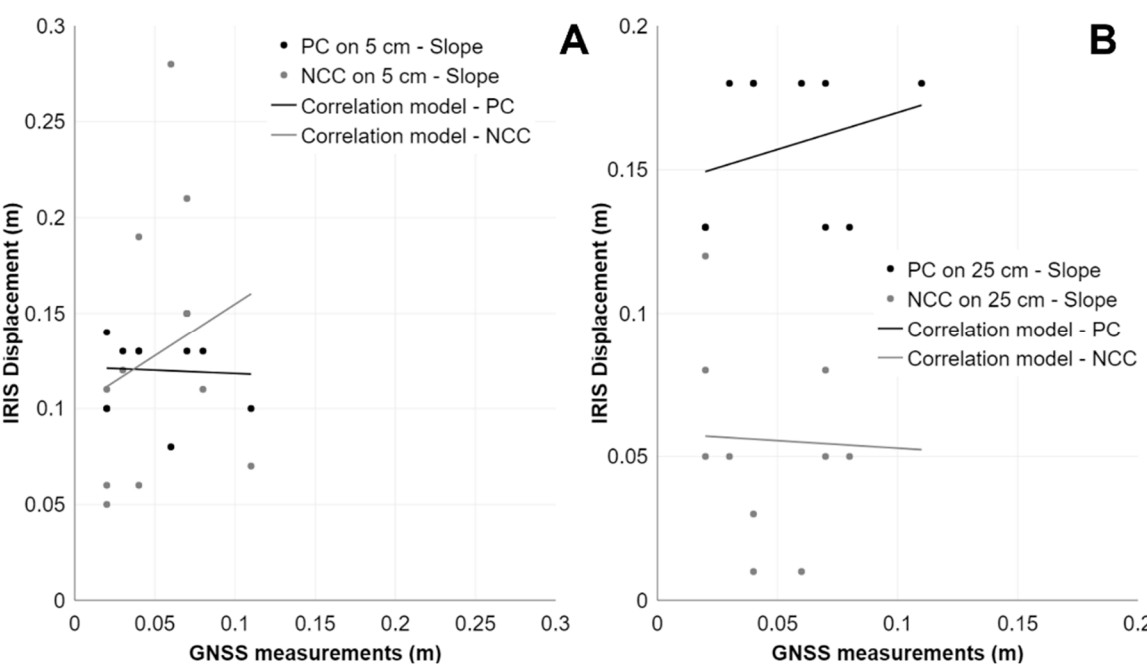

**Figure 11.** Scatter plots representing the correlation between PC and NCC applied on Trafoi SLOPE and GNSS measurements with different dataset resolutions. (**A**) PC-SLOPE-1 ($R^2$: −0.11, RMSE: 0.08); NCC-SLOPE-1 ($R^2$: −0.06, RMSE: 0.10) (**B**) PC-SLOPE-2 ($R^2$: −0.02, RMSE: 0.11); NCC-SLOPE-2 ($R^2$: −0.10, RMSE: 0.04).

## 4. Discussion

### 4.1. Explanation for the Observed Landslide Deformation Patterns

Thanks to its spatial distribution, remote sensing represents a useful tool for the analysis of geomorphic processes occurring in a study area or over a single landslide.

In the Corvara landslide, since it is characterized by some parts moving at a few meters per year [18,20], both NCC and PC, when applied to the OPH dataset (10 cm resolution) as well as the SLOPE and HSDs datasets (25 cm resolution), proved to be capable of identifying active areas at the slope scale. The quantification of movements was better with NCC and, in general, when using HSD datasets. In Ganderberg and Trafoi landslides, characterized by movements in the order of a few cm/year, by using SLOPE-1 and SLOPE-2 datasets (i.e., 5 and 25 cm resolution) and a subpixel accuracy of 0.05 px, only NCC was able to map somewhat adequately the extent of moving areas, while both NCC and PC substantially failed to adequately quantify movements of a sub-pixel magnitude.

Analysing results in more detail for the Corvara landslide, the most consistent displacement map resulted from the application of NCC on 25 cm-resolution HSD. The DIC results clearly confirm the presence of a continuous mass transfer from the source areas, converging into the track zone and slowly propagating into the accumulation zone with the development of local erosion phenomena along Rutorto river. Moreover, the material flowing from the S3 zone increases the pressure on the flat area immediately below, which is released in correspondence with the top of the T zone as rotational movements. Overall, DIC proved to be helpful in detecting the propagation of earthflow-like displacements along the entire landslide body.

The example shown in Figure 12, which represents a sector in the higher part of source zone S2, shows that the movements identified with the DIC analysis (Figure 12A) correctly correspond to true displacements recognizable by comparing Figure 12B (dated as 2019) and Figure 12C (dated as 2021). In this kind of contest, the DIC technique, and in particular NCC, is able to accurately detect the area subject to earthflow dynamics.

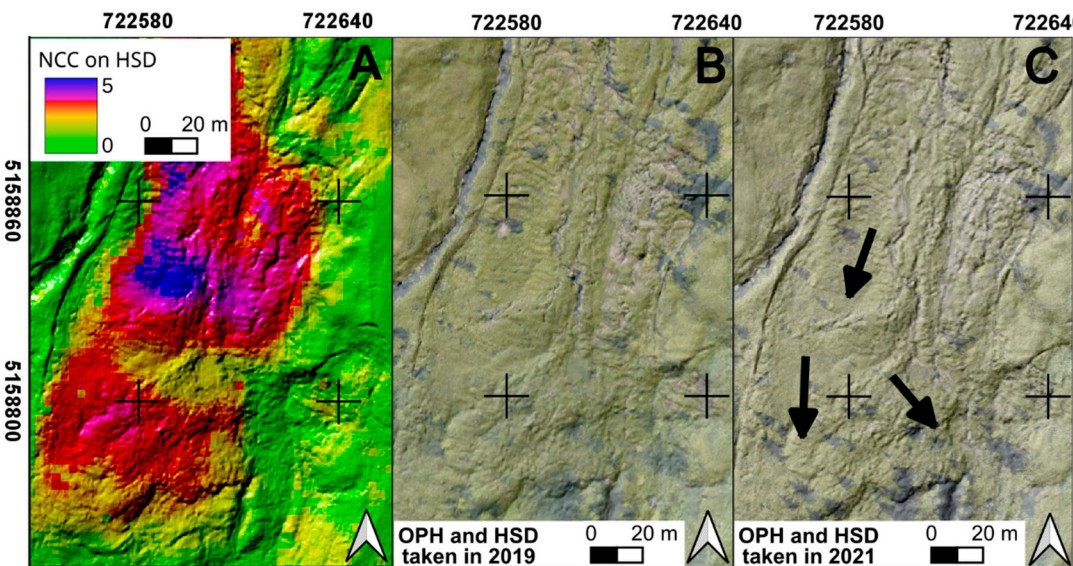

**Figure 12.** Focus on an active part of source zone S2 in the Corvara landslide. Results are compared with features on the orthophotos. (**A**) Displacement map (HSD–NCC) obtained with DIC application. (**B**) 2019-OPH overlying 2019-HSD. (**C**) 2021-OPH overlying 2021-HSD with black arrows highlighting the movement (analysis carried out using IRIS software, developed by Nhazca S.r.l.). Coordinate system: WGS84 32N.

With reference to the Ganderberg landslide, a better result for the identification of moving areas was obtained by the application of NCC on SLOPE-2 (25 cm resolution) shown in Figure 6D, while the results with the SLOPE-1 dataset (5 cm resolution), in Figure 6A,B, were very scattered and difficult to interpret. The DSGSD dynamics affecting the entire slope was not detected due to the extremely low entity of displacement; on the other hand, scree slope activity, mainly caused by weather degradation, could be recognized. Two examples of results with SLOPE-2 are presented in Figure 13, giving an overview of what the algorithm works on. Figure 13A,B represent an area near a scree slope; here, NCC searches for slope angles contrasts that correspond to the alternation of greyscale colours that leads the algorithm to clearly identify movements but is not able to quantify it properly. Figure 13C,D represent an area of the national road SS 44bis, north of the little village of Hahnebaum, where the active parts identified by DIC are correlated with more continuous patches of SLOPE-2. In this case, NCC finds coherent intensity differences more easily, which mostly correspond to true displacements.

The most consistent result for the Trafoi landslide site (likewise in the Ganderberg case) is obtained from the application of NCC on the SLOPE-2 dataset, shown in Figure 9D. However, contrary to the Ganderberg case, the Trafoi site results both help to discriminate between stable and unstable areas and recognize movements within the scree slopes. In fact, looking again at Figure 9D, we can identify a general movement of the active rockslide (Unit A), while no significant displacement is recognized in the lower part, which corresponds to the evolved and stable rockslide (Unit B). In Figure 14, a focus on the investigated features of the slope map is presented. Figure 14A shows a part of the central sector of the rockslide, where a general south-east displacement is recognized. Within this sector, NCC calculated some movements peaks, in red patches, that can be attributed to features highlighted in Figure 14B,C. Usually, artefacts, which are due to a DTM not properly cleared from noise such as vegetation, lead NCC to this kind of erroneous matches (Figure 14B). On the other hand, when artefacts are not present, NCC does not always find the true correspondence between contrasts as the slope band shift during the two-year period is limited, thus leading to mismatches (Figure 14C).

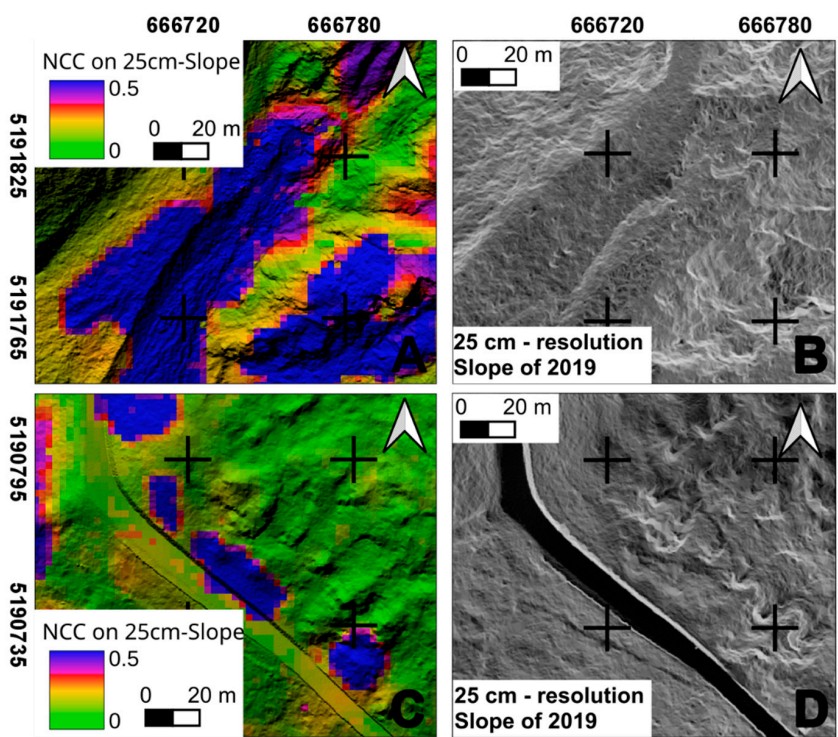

**Figure 13.** Examples of computed active parts in the Ganderberg landslide. Results are compared with features on the SLOPE-2 dataset. (**A**,**B**) refer to scree slopes, while (**C**,**D**) refer to a part of the national road, north of Hahnebaum (analysis carried out using IRIS software, developed by Nhazca S.r.l.). Coordinate system: WGS84 32N.

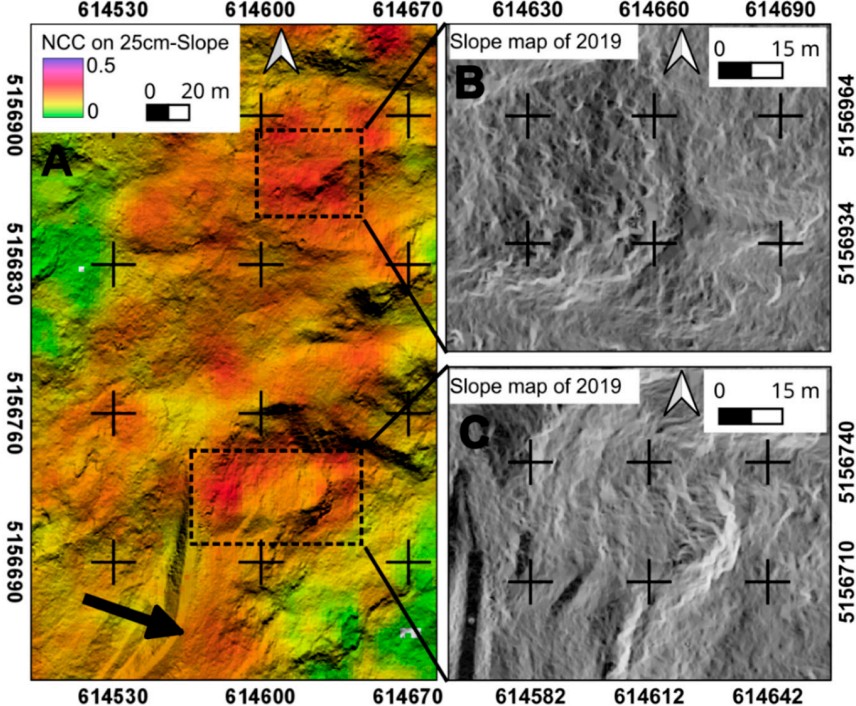

**Figure 14.** Examples of computed active parts in the Trafoi landslide. Results are compared with features on the SLOPE-2 dataset. (**A**) Displacement map (SLOPE–NCC) obtained with DIC application. The black arrow indicates the general south-east movement of rockslide. (**B**,**C**) show the features on which the NCC algorithm works in the slope map (analysis carried out using IRIS software, developed by Nhazca S.r.l.). Coordinate system: WGS84 32N.

## *4.2. Performances of PC and NCC in the Recognition of Slope Movements on a Qualitative Basis*

Regarding the performance of PC and NCC in the recognition of slope movements on a qualitative basis, PC seems to perform better than NCC with high movement rates, while NCC seems to perform relatively better than PC with low movement rates. The limitation of PC in identifying small sub-pixel movements is likely related to the fact that it computes displacement based on the dominant phase difference. Therefore, if the morphology and coverage of the slope are such that movements are evidenced by a few pixels only, then the phase differences associated with these few pixels are ignored, while phase differences that cover all frequencies (which might be related to lower sampling of ground points due to vegetation or unfavourable slope conditions) lead to an inaccurate determination of the dominant phase peak—that is, to an inaccurate displacement computation. In addition, PC is more sensitive to data georeferencing, which might cause small misalignments between multi-temporal images. In such a case, given the limited amount of real movements, the algorithm struggles to find an agreement between phase differences in different frequencies within the master and slave images [31]. These are probably among the reasons why PC shows scattered pattern of movement values in Ganderberg and Trafoi landslides, especially with high-resolution data (i.e., SLOPE-1 and 5 cm pixel size) which are affected by dense vegetation and rock blocks spread along all the landslide bodies, and in which imperfect georeferencing can mask actual movements. On the other hand, the NCC method tracks sharp differences in pixel values, such as those due to colour contrasts between rocks and grass in orthophotos or passages from steep to flat slopes in DTM-derived slope maps. These differences must be found in both master and slave images, and their displacements must correspond to the true landslide movement; if not, NCC generates erroneous matches between features [31]. This requirement is satisfied in the Corvara landslide, especially with HSD and SLOPE maps, where the faster earthflow parts in the source and track zones (S2, S3, and T zone) preserve these contrasts along with the movements. On the other hand, the NCC can recognize the general movement of the rockslide unit (Unit A) even in the Trafoi landslide, as well as along the national road and the scree-slopes in the Ganderberg landslide, thanks to the fact that abrupt changes in slope angle determine sharp differences in pixel values which can be tracked by NCC.

## *4.3. Performances of PC and NCC Algorithms to Assess Displacement on a Quantitative Basis*

Regarding the performance of the PC and NCC algorithms to assess displacement on a quantitative basis, the results show that the smaller the real movements, the larger in percentage the mismatch is between the DIC-computed and real displacements, and that a better quantification is obtained with DTM-derived datasets (SLOPE and HSD). In the Corvara landslide, locally characterized by moderate velocities, the PC- and NCC-computed displacements show some decent correlation with the validation datasets (HPT and GNSS). Looking at the HPT validation plots and the values of $R^2$ (adjusted R-squared) and RMSE (Root Mean Squared Error), the best combinations for the S2 and T zones are obtained with the application of NCC on 25 cm HSD (Figure 4B,E), with an $R^2$ of 0.41–0.87 and an RMSE of 0.83–0.61. Looking at the GNSS validation plots, instead, the best combination is obtained with NCC on 25 cm SLOPE, with an $R^2$ of 0.36 an RMSE of 0.36 (Figure 5C). The applied validation methods proved that, dealing with a phenomenon associated with a significant cumulative displacement (at least 5 m over a two-year period), the DIC technique, analysing HSD and SLOPE at 25 cm resolution, was capable of detecting and quantifying movements with a sufficient spatial continuity and accuracy. In the Ganderberg landslide site, characterized by slow to extremely slow movements, the correspondence between NCC- and PC-computed displacements and HPT as well as GNSS measurements is rather poor. Looking at the HPT analysis, it appears that the result given by PC applied on SLOPE-2 (25 cm resolution) datasets ($R^2$ of 0.20 and RMSE of 0.31) is slightly better than that obtained with NCC ($R^2$ of 0.14 and RMSE of 0.29) on the same datasets (Figure 7B); this is probably due to the two outliers in the NCC plot, which compute more than 1 m of displacement on the upper part of the plot. In fact, comparing HPT validation with

displacement maps in Figure 6, the NCC map (Figure 6D) is still much smoother than the PC one (Figure 6C). However, no significant correlations are highlighted with GNSS data. Finally, in the Trafoi site, another slow to extremely slow landslide, there is no model that leads to an adequate quantitative agreement between computed displacements and HPT analysis or GNSS validation datasets. Looking at the HPT validation, the best combination between $R^2$ (−0.001 to −0.002) and RMSE (0.32 to 0.34) values with the highest precision, but also the worst accuracy, is represented by both NCC and PC applied on the SLOPE-2 dataset (25 cm resolution) (Figure 10B). Even in this case, no significant matches were highlighted by the GNSS validation method.

## 5. Conclusions

The results obtained in this study for the detection of the recent dynamics in large-scale landslides via DIC of airborne optic and LiDAR datasets in three different test sites in South Tyrol demonstrate that DIC's application works quite well for the identification and quantification of movements in a multi-pixel range of magnitude and in non-densely vegetated areas. However, this approach encounters significant difficulties in identifying and quantifying movements which are in a sub-pixel range of magnitude, particularly in vegetated areas in which both types of datasets lose the capacity to depict slope features adequately.

As a matter of fact, in the Corvara landslide, where movements were up to few meters during the 2019–2021 period, the DIC application proved to be a powerful tool to complement other monitoring technique in fast-changing landslides environments, giving a valuable insight of the overall development patterns of the phenomenon at the slope scale. On the other hand, in the Ganderberg and Trafoi slow-moving landslides, where GNSS benchmarks during the two-year period showed that cumulative displacement was limited to 11–16 cm, neither NCC nor PC in any DTM dataset was able to provide detailed information about slope movements, since they are significantly influenced by noise and thus compute flawed matches, which inevitably lead to inaccurate estimates of displacements. Even the high-resolution 5 cm datasets could not provide a better result. Slow-moving landslides are in fact characterised by such low cumulative displacement that by analysing images with more details the number of features that could cause mismatches increases. This highlights that the theoretical sub-pixel accuracy of DIC techniques cannot be considered practically achievable in such challenging conditions and that such techniques should be applied over longer periods of time, to reach a properly detectable amount of cumulative displacement.

In conclusion, in all case studies, DIC algorithms proved to be adequate, at least for differentiating at the slope scale the active parts from the stable ones and, in one case study, assessing movements quantitatively. These results are of significant added value for targeting punctual in situ monitoring from the perspective of the long-term surveillance of large-scale landslides. Once more, we stress the important role that spatially distributed DIC results play in the comprehension and interpretation of geomorphic dynamics. In this paper, multiple applications of the DIC methodology for landslide displacement quantitative estimation are given: the analysis can be performed on different remote sensing products such as hillshade and slope maps derived from high-resolution LiDAR or optical scenes using multiple algorithms (NCC or PC). Our work adds to previous studies where DIC has been applied to airborne and satellite acquisitions [16,26,37]. Moreover, the tuning of DIC elaboration and subsequent results can be improved when local pointwise displacement information (GNSS or other ground-based instruments) is present for the study area.

**Author Contributions:** SoLoMon project responsibility and management: V.M. and D.T.; External scientific units' responsibility: A.C. and G.M.; Data processing and analysis: M.T., M.M., G.C. and A.C.; Results analysis and discussion: M.T., M.M., G.M., G.B. and A.C.; Manuscript preparation (text, figures, and tables): M.T., G.B. and A.C.; Internal review and editing: M.T., M.M., G.C., G.M., G.B., D.T., V.M. and A.C. All authors have read and agreed to the published version of the manuscript.

**Funding:** This research was funded by the SoLoMon project "Monitoraggio a Lungo Termine di Grandi Frane basato su Sistemi Integrati di Sensori e Reti" (Long-term monitoring of large-scale landslides based on integrated systems of sensors and networks), Program FESR 2014–2020, Project FESR4008 South Tyrol–Responsible. V. Mair.

**Data Availability Statement:** The data presented in this study are owned by the Autonomous Province of Bolzano, the University of Modena and Reggio Emilia and the National Research Council, and they are not available publicly.

**Acknowledgments:** Nhazca S.r.l. is acknowledged for providing the University of Modena and Reggio Emilia with IRIS Software for scientific application and testing purposes. The Forestry Department of the Bolzano Autonomous Province is acknowledged for providing periodic GNSS surveys data.

**Conflicts of Interest:** The authors declare no conflict of interest.

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
