# Peer review of "Detecting Recent Dynamics in Large-Scale Landslides via the Digital Image Correlation of Airborne Optic and LiDAR Datasets: Test Sites in South Tyrol (Italy)"

_remotesensing, doi:10.3390/rs15122971_

Round 1
Reviewer 1 Report
In this paper, the authors tried to test the possibility to retrieve significant slope displacement data from the analysis of multi-temporal airborne Optic and Light Detection and Ranging (LiDAR) surveys with Digital Image Correlation (DIC) algorithms such as Normalized Cross-Correlation (NCC) and Phase Correlation (PC). The topic falls into the scope of Remote Sensing. Some suggestions for the authors:
1. How to obtain the optical camera and the LiDAR point clouds data by UAV, which should be mentioned in this paper.
2. Are the time intervals in the three data acquisition processes the same? Or whether the date of data collection each year is the same or adjacent.
3. In Section 2.3 of Methods,Why the grayscale OPH datasets is used in this paper.
4. In Table 4, why the Fast static technique is used when collecting GNSS data for Corvara landslide, but the combination of Fast static and RTK is used for Ganderberg landslide and Trafoi landslide.
5. Why there are two gray lines in Figure 3.f and Figure 6.d.
6. In Figure 3.h, the image in the top left corner is not so clear (especially the text).
7. In line 377, maybe it should be 3.3. Trafoi landside.
8. Adjust the position of Figure 3, Figure 6 and Figure 9 appropriately, and promise them to be in the middle.
Author Response
Please see the attachment.
Kind regards
Melissa Tondo, on behalf of the other authors

Reviewer 2 Report
In my opinion it is a necessary research on the detection of ground movement of large scale using remote sensing methods like LiDAR; especially when it is compared with ground movement using GNSS techniques.
The description, analysis and results are presented in a proper order and with a proper details.
There are a few changes required with respect to the language - they are highlighted in yellow in the attached file.

There are a few changes required with respect to the language - they are highlighted in yellow in the attached file.
Author Response

(The authors gave the same response as above.)

Reviewer 3 Report
Dear authors,
after reviewing the paper, I suggest that the manuscript should be accepted after minor revision.
In general
The presented topic is interesting and up to date, considering the datasets and methods used for the analyses, as well as their application to landslides as one of the most serious natural hazards. The paper is well structured with all sections included, with clearly separated methodology, results, and discussion.
The main comment is related to the presentation of the results in the text, where in my opinion all the values are not supported by those presented in graphs. Please, check the values and/or better explain.
> Lines 285-288 – (i) it is stated that “calculated with PC often reach values of 4-5 m”, but looking at the graphs A, B, and C, only a few points are above the value 4 calculated with PC; (ii) it is stated that “calculated with NCC range mainly up to 1.5-3 m”, but looking at the graph A there are no points for NCC above 1.5 m and looking at the graph B there are some points for NCC even above the value of 3 m.
> Lines 295-296 – (i) it is stated that “GNSS measurements range from 0.5 to 0.7 m”, but despite it’s hard to read the exact values on the graphs, the values range from 0 to the value lower than 0.7; (ii) it is stated that “DIC results reach up to 1.2-1.3 m”, but looking at the graphs, the values are dominantly below 1.2 m.
> Lines 433-434 – It is stated that “Resulted better with PC”, but earlier in the text (line 291) it is stated that “the best correlations are obtained with NCC application”, which is also presented in Figures 4B and 4E.
Figures & Tables
> Figure 1 – I suggest doing the following: (i) increase the text font in Figures B, C, and D; (ii) increase the width of the zone lines, (iii) add an arrow of the general movement direction; (iv) to be easier to follow, I suggest naming the zones in Figure 1B as S2, S3, T and A (the same as in Figure 3, and in the further text, e.g. line 274).
> Figure 3, Figure 6, Figure 9 – I suggest doing the following: (i) increase the text font; (ii) increase the width of the zone lines.
> Table 1 – I suggest doing the following: (i) remove all “,” in the column “Period”; (ii) in the column “Original data” replace “full res.” with “fr-” (e.g., full res. OPH change to fr-OPH), so that it would be equal as in the text above (lines 162-164).
> Table 3 – I suggest removing all “px” within the Table and putting “(px)” in the Name of the column (e.g. “Search window (px)”)
> Caption for Figure 3 – The caption is the repetition of the text from the beginning of the subsection (lines 263-272), where a detailed explanation is given. Thus, I suggest using the abbreviations used in the text, and changing the caption to “Displacement maps related to the evolution of the Corvara landslide during the 2019 – 2021 period: a) PC-OPH; b) NCC-OPH; c)…” Leave the description for g) and h). Correct the description for f) where the slope map is presented, not the hilshade.
> Captions for Figure 6 and Figure 9 – The same comment as for Figure 3, explained above.
> Captions for Figures 3, 6, and 9 – As it is several times in the text already stated that analyses are performed using IRIS software developed by Nhazca s.r.l. (e.g. lines 194-195, Caption of Figure 2, etc.), it is not necessary to repeat this also within these captions.
References
> As many authors presented the application of DIC for landslide investigation, I suggest adding in the Introduction chapter (e.g. line 63) that DIC has also been recognized as a powerful method in landslide monitoring, using various input datasets from different platforms (e.g. Bickel et al., 2018; Caporossi et al., 2018; Mulas et al., 2020). If the authors accept this suggestion, accordingly, the number of references must be changed in the References and the text.
Bickel et al., 2018, Quantitative Assessment of Digital Image Correlation Methods to Detect and Monitor Surface Displacements of Large Slope Instabilities, Remote Sensing, 10(6), 865.
Caporossi et al., 2018, Digital Image Correlation (DIC) Analysis of the 3 December 2013 Montescaglioso Landslide (Basilicata, Southern Italy): Results from a Multi-Dataset Investigation, ISPRS International Journal of Geo-Information, 7(9), 372.
Mulas et al., 2020, Integration of Digital Image Correlation of Sentinel-2 Data and Continuous GNSS for Long-Term Slope Movements Monitoring in Moderately Rapid Landslides, Remote Sensing, 12(16), 2605.
Minor corrections and suggestions
> Consider replacing the term “optic” with “optical”.
> Line 29 – Remove space before “DIC”.
> Line 57 – As for the first time the SAR abbreviation is used, add “synthetic aperture radar (SAR)”.
> Line 69 – Leave in the text “and” or “etc”.
> Lines 81, 288, 289, 292, 298, 340, 341, 344, 345, 371, 372, 399, 402, 405, 406, 423, 424, 425, 443, 472, 474, 524, 527, 537, 539, 540, 546 – Replace “Fig.” with “Figure” according to Instructions for Authors.
> Lines 88, 104, 134 – Usually, the range is expressed from a smaller to a larger number. Not necessarily, but I suggest changing it to “1550 m to 2080 m”, “1170 m to 2330 m”, and “1500 m to 2700 m”.
> Line 88 – Remove “,” behind “100 m”.
> The sentence that starts in line 117 (“Periodic GNSS…”) is not finished, please join with the next one.
> Line 137 – Replace “:” with “.” “Unit A…” is a new sentence.
> Line 157 – Add “,” after “2020”.
> Line 158 – Remove “-“ after “snow”.
> Lines 161, 205, 237, 245 – Replace “Tab.” with “Table” according to Instructions for Authors.
> Lines 162-163 – Equalize “full-resolution” and “full resolution”.
> Lines 166, 195, 202, 307, 354, 397, 452, 516, 522 – Change “S.r.l.” to “s.r.l.”.
> Line 193 – Add the abbreviation “FFT” after “Fast Fourier transforms”, as it is used later in Figure 2.
> Line 207 – It is not clear why is stated “lowered/full-resolution” for HSD when only one resolution is used (according to Table 2).
> Line 272 – As it is already used before (line 248), leave in the text only the abbreviation “HGF”.
> Line 274 – Add ”,” after “S3 zone”.
Kind regards
Author Response

(The authors gave the same response as above.)

Reviewer 4 Report
In this paper, three large-scale landslides have been selected for testing the possibility to retrieve significant slope displacements from the analysis of multi-temporal airborne LiDAR surveys with Digital Image Correlation (DIC) algorithms such as Normalized Cross-Correlation (NCC) and Phase Correlation (PC). The results suggest that DIC works quite well for the identification and quantification of movements in a multi-pixel range of magnitude and in non-densely vegetated areas. The following are some comments for a possible improvement of this manuscript.
1. In the Introduction, there is no literature review of previous relevant studies. The authors should improve the elaboration of scientific (or technical) problems.
References:
Pan, B., Xie, H. M., Xia, Y., & Wang, Q. (2009). Large deformation measurement based on reliable initial guess in digital image correlation method. Acta Optica Sinica, 29(2), 400-406.
Daehne, A., & Corsini, A. (2013). Kinematics of active earthflows revealed by digital image correlation and DEM subtraction techniques applied to multi‐temporal LiDAR data. Earth Surface Processes and Landforms, 38(6), 640-654.
Okyay, U., Telling, J., Glennie, C. L., & Dietrich, W. E. (2019). Airborne lidar change detection: An overview of Earth sciences applications. Earth-Science Reviews, 198, 102929.
2. In the Methods, the principle of the method is not described in details. Relevant formulas or images can be added to make people understand it.
3. In the Discussion, there is no detailed analysis and explanation of the deformation results. Although this paper is more like a “technical note”, readers may be also interested in some explanations for the observed landslide deformation patterns.
References:
Meng, Y., Lan, H., Li, L., Wu, Y., & Li, Q. (2015). Characteristics of surface deformation detected by X-band SAR Interferometry over Sichuan-Tibet grid connection project area, China. Remote Sensing, 7(9), 12265-12281.
4. The significance of this paper is not elaborated sufficiently. The authors need to highlight the innovative contribution of this paper.
Author Response

(The authors gave the same response as above.)
